# A Mixed-Lipid Emulsion Containing Fish Oil for the Parenteral Nutrition of Preterm Infants: No Impact on Visual Neuronal Conduction

**DOI:** 10.3390/nu13124241

**Published:** 2021-11-25

**Authors:** Christoph Binder, Hannah Schned, Nicholas Longford, Eva Schwindt, Margarita Thanhaeuser, Alexandra Thajer, Katharina Goeral, Matteo Tardelli, David Berry, Lukas Wisgrill, David Seki, Angelika Berger, Katrin Klebermass-Schrehof, Andreas Repa, Vito Giordano

**Affiliations:** 1Comprehensive Center for Pediatrics, Department of Pediatrics and Adolescent Medicine, Division of Neonatology, Intensive Care and Neuropediatrics, Medical University of Vienna, 1090 Vienna, Austria; christoph.a.binder@muv.ac.at (C.B.); hannah.schned@muv.ac.at (H.S.); eva.schwindt@muv.ac.at (E.S.); margarita.thanhaeuser@meduniwien.ac.at (M.T.); alexandra.thajer@muv.ac.at (A.T.); katharina.goeral@muv.ac.at (K.G.); lukas.wisgrill@muv.ac.at (L.W.); angelika.berger@muv.ac.at (A.B.); katrin.klebermass-schrehof@meduniwien.ac.at (K.K.-S.); vito.giordano@muv.ac.at (V.G.); 2Neonatal Data Analysis Unit, Department of Medicine, Chelsea and Westminster Campus, School of Public Health, Imperial College London, London SW10 9NH, UK; n.longford@imperial.ac.uk; 3Department of Medicine, Division of Gastroenterology and Hepatology, Weill Cornell Medical College, New York, NY 10065, USA; matteotardelli1@gmail.com; 4Centre for Microbiology and Environmental Systems Science, Department of Microbiology and Ecosystem Science, Division of Microbial Ecology, University of Vienna, 1090 Vienna, Austria; david.berry@univie.ac.at (D.B.); sekifilipdavid@gmail.com (D.S.)

**Keywords:** fish oil, omega-3 fatty acid, lipid emulsion, parenteral nutrition, brain maturation, visual evoked potential, visual neuronal conduction

## Abstract

Fish oil is rich in omega-3 fatty acids and essential for neuronal myelination and maturation. The aim of this study was to investigate whether the use of a mixed-lipid emulsion composed of soybean oil, medium-chain triglycerides, olive oil, and fish oil (SMOF-LE) compared to a pure soybean oil-based lipid emulsion (S-LE) for parenteral nutrition had an impact on neuronal conduction in preterm infants. This study is a retrospective matched cohort study comparing preterm infants <1000 g who received SMOF-LE in comparison to S-LE for parenteral nutrition. Visual evoked potentials (VEPs) were assessed longitudinally from birth until discharge. The latencies of the evoked peaks N2 and P2 were analyzed. The analysis included 76 infants (SMOF-LE: *n* = 41 and S-LE: *n* = 35) with 344 VEP measurements (SMOF-LE: *n*= 191 and S-LE *n* = 153). Values of N2 and P2 were not significantly different between the SMOF-LE and S-LE groups. A possible better treatment effect in the SMOF-LE group was seen as a trend toward a shorter latency, indicating faster neural conduction at around term-equivalent age. Prospective trials and follow-up studies are necessary in order to evaluate the potential positive effect of SMOF-LE on neuronal conduction and visual pathway maturation.

## 1. Introduction

Preterm infants, especially those who are born with an extremely low birth weight (ELBW, <1000 g), are at an increased risk of abnormal brain maturation and development [1]. Omega-3 long-chain polyunsaturated fatty acids (LC-PUFA) are essential constituents of biological membranes and contribute to maintaining the structural and functional integrity of cellular components [2,3]. Omega-3 LC-PUFA docosahexaenoic acid (DHA) is highly concentrated in neuronal membranes and retinal ganglion cells and is crucial for normal brain maturation [2,4,5]. Preterm infants are typically deficient in DHA for several reasons, including the following circumstances: (1) DHA accretion occurs during the last trimester of pregnancy; (2) DHA supply from enteral nutrition falls short of fetal demands; and (3) the capacity to synthesize DHA from precursor cells is very low [6,7]. Preterm infants depend on parenteral nutrition for several weeks and the standard soybean oil-based lipid emulsion (S-LE) for parenteral nutrition is almost devoid of DHA [8]. A mixed-lipid emulsion composed of soybean oil, median-chain triglycerides, olive oil, and fish oil (SMOF-LE) provides DHA and its precursor eicosapentaenoic acid (EPA) [9]. A previous study demonstrated that electrophysiological brain maturation, as measured by amplitude-integrated electroencephalography, was accelerated in infants who received a mixed-lipid emulsion for parenteral nutrition [10]. The potentially positive effects on neuronal conduction, as measured by visual evoked potentials, have not been investigated so far. Visual evoked potentials (VEPs) are electrophysiological signals generated in response to visual stimulation [11]. In preterm infants, VEPs are primarily used to assess visual neuronal conduction and the maturation of the visual pathway [12]. VEPs consist of three principal peaks (P1, N2, P2) that can be recorded from approximately 23 weeks of gestation and show an ongoing maturational process in terms of both velocity and morphology in the postnatal period [12,13]. In fact, VEP latencies become faster with increased gestational age, and having late component peaks such as N2 and P2 is more stable over the time compared to having early component peaks such as P1 that have been shown to be less reliable [12,14]. The aim of this study was to evaluate the effect of SMOF-LE in comparison to a pure soybean oil-based lipid emulsion (S-LE) on visual neuronal conduction, as measured by serial visual evoked potentials from birth until discharge.

## 2. Materials and Methods

### 2.1. Study Design

This retrospective matched cohort study was conducted at the Department of Paediatrics and Adolescent Medicine at the Medical University of Vienna between the years 2009 and 2019. The study was approved by the local ethics committee (Nr. 2148/2020). In order to detect potential differences in preterm infants’ brain maturation, we investigated visual neural conduction measured by serial VEPs in infants who received SMOF-LE in comparison to S-LE for parenteral nutrition.

### 2.2. Patient Groups

Infants born with an extremely low birth weight (<1000 g) between the years 2009 and 2019 who received serial VEP measurements from birth until discharge were included in this study analysis. Infants in the two study groups were matched (1:1) for sex, gestational age at birth (+/−3 days), and birth weight (+/−100 g) [15,16].

This study compared two different parenteral lipid emulsion periods: Infants in the SMOF-LE group received a mixed-lipid emulsion (SMOF-LE, SMOFlipid^®^ 20%; Fresenius Kabi), which contained 30% soybean oil, 30% medium-chain triglycerides, 25% olive oil, and 15% fish oil (2015–2019), while infants in the S-LE received a pure soybean oil-based lipid emulsion (S-LE, Intralipid^®^ 20%; Fresenius Kabi) (2009–2012) for parenteral nutrition [17]. Data referred to two different epochs where two different lipid emulsions (S-LE vs. SMOF-LE) were used in clinical practice and serial VEP measurements were performed from birth until term-equivalent age as part of two observational studies [18,19]. S-LE was the standard lipid emulsion in our unit before the year 2012; after a randomized controlled trial [20] on parenteral nutrition (2012–2015) we switched to SMOF-LE for parenteral nutrition. SMOF-LE contains 2.2% DHA and 2.4% EPA, while S-LE is devoid of DHA and EPA [17]. Enteral nutrition was commenced using breast milk or donor milk (provided from June 2012) and increased gradually according to the local protocol (maximum of 20 mL/kg per day). If donor milk was used, infants were switched to preterm formula after 32 weeks of postmenstrual age (PMA). Milk feedings were fortified when the level of 100 mL/kg of enteral nutrition was reached. The standard local nutritional protocol for parenteral nutrition did not differ substantially between the two nutritional eras, except for the type of lipid emulsion (S-LE and SMOF-LE). Infants received full parenteral nutrition directly after birth and lipid emulsions were started at 1 g/kg/d. Lipids were increased daily by 0.5 g/kg/d and dosed up to 3.5 g/kg/day, at the discretion of the attending physicians, and reduced in relation to enteral nutrition (increased up to 20 mL/kg/day). Parenteral nutrition was stopped at 140–160 mL/kg/d of enteral nutrition. Due to predicted negative outcomes on VEPs, patient data reporting severe retinopathy of prematurity (grade > II), cystic periventricular leukomalacia, and severe intraventricular haemorrhage (grade > II) were not included in this study [21,22,23]. Parenteral nutrition-associated cholestasis, abdominal surgery for necrotizing enterocolitis or focal intestinal perforation [23], and genetic or metabolic disorders were exclusion criteria. VEP measurements with impedance over 10 kΩ and artefacts were also excluded from the analysis [18].

### 2.3. Visual Evoked Potentials

Visual evoked potentials were analyzed longitudinally (maximum 10 days between consecutive measurement) from birth until discharge using flash VEPs. Data on serial VEP measurements were obtained from two independent studies and retrospectively analyzed [18,19]. The first VEPs were performed as soon as the clinical condition of the preterm was stable. Measurements were assessed using a Nihon Kohden MEB-9400 Neuropack S1 device. Surface gold electrodes were positioned at Oz(+) and at Fz(−) according to the international 10/20 system. Red light LED goggles were used with 0.4 cd/m^2^ and held at a distance of 5 cm in front of the infant’s eyes [18]. Impedance was kept below 10 kΩ, the stimulation frequency was 0.7 Hz, the bandpass filter was 1–100 Hz, and the sweep time was 1 s. The procedure was performed in a semi-dark environment. At least two repetitions of 30 averages were obtained from both eyes. A total mean repetition curve was obtained from every single repetition considered and the consecutively most stable peaks (P1, N2, P2) were considered in the statistical analysis (Figure 1) [11,14]. VEP peaks refer to retino-geniculo-cortical pathway activation and their latencies and morphology changes with increased GA [12,13]. A precise origin of P1 has not been identified yet. P1 usually appears between the 28th and the 32nd gestational week and could be the result of basal dendrite activities [13,24]. Even if its presence is an index of visual development, it has been described as a less reliable peak for use in a population of premature infants [18,24]. Early-phase late VEP components (N2) are more likely to be generated in the dorsal extrastriate cortex of the middle occipital gyrus, while the late components (P2) refer to near associative areas [25].

### 2.4. Statistical Analysis

Data were analyzed with IBM SPSS Statistics^®^ (International Business Machines Corporation, Statistical Package for the Social Sciences) (IBM Corp., Armonk, New York, NY, USA; Version 26) and the R statistical software (R Foundation for Statistical Computing, Vienna, Austria; Version 3.6.2). For metric parameters, means and standard deviations (SD) were calculated, while for nominal and ordinal parameters absolute frequencies were specified. A two-sample *t*-test with a two-sided significance level of α = 0.05 was used for calculating and comparing baseline characteristics as well as VEP values, while for the categorical data a χ^2^-test with a two-sided significance level of α = 0.05 was applied.

The analysis of the three outcome variables P1, N2, and P2 comprised three steps: (1) the prediction of the outcomes at the set postmenstrual ages (PMA); (2) propensity balancing; and (3) the analysis of the outcomes with the inverse propensity weights. These steps were applied separately to P1, N2, and P2 and for the distinct PMAs.

#### 2.4.1. Prediction of the Outcomes

The three outcome variables, P1, N2m and P2, were analyzed for an infant at up to nine time points: 28, 29, 30, 31, 32, 33, 34, 35, and 36 weeks PMA. The values were estimated from the available observations by normal kernel smoothing [26].

#### 2.4.2. Propensity Balancing Analysis

For the observations included in the analysis, we set the balancing weights so as to match the weighted means of the groups for the background variables of sex (male/female), gestational age at birth (+/−3 days), and birth weight (+/−100 g) [15,16]. The balance was described by the three scaled differences of the within-group means. The scaling was carried out using the pooled SD of the variable. These balances were the sole diagnostic, or assessment, of the quality of the balancing. They were compared to the scaled differences evaluated without the balancing weights. It was essential for the balancing to be based only on the background variables (and the treatment) and not be informed by the values of the outcomes except, indirectly, by the PMAs at which the observations were taken.

#### 2.4.3. Analysis of the Outcomes

The hypothesis that the outcomes have identical means in the two groups was tested using the t statistic adapted for the balancing of the weights. By way of a sensitivity analysis, we repeated the analyses (prediction, balancing, and comparison of the outcomes) for values of PMA from 28 to 36 weeks. In normal kernel smoothing, the value of the smoothed fit at a given time point was defined as the weighted mean of observations. The weights were set according to the distance of the observation times and the evaluation time point. For an observation made at time point t, the weight used for estimating the values at time point s was defined as the density of the standard normal distribution at (*t*-s)/sigma, where sigma was the SD of the kernel. After exploring several alternatives, we chose sigma = 10 (days). That is, the weight of an observation at time point s (*t* = s) is 1.00, at *t* = s + 11.75 and *t* = s − 11.75 is 0.50, and at *t* = s + 16.60 and *t* = s − 16.60 is 0.25. The precision of the prediction at a point s is summarized by the total of the weights of the infants’ observations. From an analysis of values at a particular PMA (s), we excluded all infants with the total kernel weight smaller than a given threshold (0.10).

## 3. Results

A total number of 118 preterm infants with longitudinal VEP measurements from birth until discharge were screened for the study analysis. After applying the exclusion criteria, 76 infants (SMOF-LE: *n* = 41 and S-LE: *n* = 35) with 344 VEP measurements (SMOF-LE: *n*= 191 and S-LE *n* = 153) were included in the analysis (Figure 2). Groups were well matched, and no significant differences were found in the descriptive characteristics, co-morbidities, or nutritional aspects (Table 1). The type of feeding at discharge, days on parenteral nutrition, and parenteral lipids were not significantly different between the two groups (Table 1).

Considering the strong maturational feature of P1, which is consistent around term-equivalent age [18], and having less observations for this peak, it was not possible to perform a predictive value analysis in this case.

A maturational feature could be observed for both considered peaks: N2 and P2. Precise latency values for these peaks at different PMAs are reported in Figure 3 (unadjusted values). As expected, the N2 and P2 latencies decreased continuously from birth until discharge in both groups. Starting at 34 weeks PMA, the values of N2 and P2 were faster in the SMOF-LE group in comparison to the S-LE group but did not reach statistical significance (Figure 3).

The treatment effect between the two groups was calculated and showed a trend towards a shorter latency of VEPs in the SMOF-LE group at around term-equivalent age but did not meet statistical significance (Figure 4 and Table 2).

## 4. Discussion

Visual neuronal conduction, as measured by VEPs, was not significantly different in preterm infants who received a mixed-lipid emulsion containing fish oil (SMOF-LE) in comparison to a soybean oil-based lipid emulsion (S-LE) for parenteral nutrition. Starting at 34 weeks PMA, the treatment effect showed a trend towards accelerated visual neuronal conduction in the SMOF-LE group in comparison to the S-LE group but did not reach statistical significance.

In vitro studies have shown that DHA supplementation uniquely promotes neurite growth and synaptic activity [27,28,29]. In a previous study, we found that electrophysiological brain maturation, as measured by amplitude-integrated EEG, was accelerated in infants receiving SMOF-LE in comparison to S-LE [10]. The potentially beneficial effect of SMOF-LE on neuronal conduction, as measured by VEPs, has not been investigated yet. VEPs measure the functional integrity of the visual pathways from the retina via the optic nerves to the visual cortex and correlate with the synaptic activity and visual pathway maturation [25]. Thus, the use SMOF-LE for parenteral nutrition might have a positive effect on visual neuronal conduction and may have caused the accelerated maturation of the visual pathway in our study.

Several studies have evaluated the effect of maternal LC-PUFA supplementation during pregnancy and lactation on visual evoked potential in term infants, with conflicting results [30,31,32,33,34]. A study by Bauer et al. [34] demonstrated that LC-PUFA supplementation in adults has a positive effect on cortical visual processing, including improving peak amplitudes and reducing latencies. However, this was associated with EPA supplementation and not with DHA. While DHA seems to play a very important role during the first years of development, EPA is relevant in adults [2]. In particular, EPA is considered a biomarker for pre-dementia syndrome and cognitive decline [2]. However, fish oil is rich in DHA and EPA, and analysis of the individual omega-3 fatty acids is almost impossible.

As in previous studies, we observed a maturational feature for the VEP components and therefore a reduction in latencies for both N2 and P2. Values of around 300 ms were observed for N2, while values of around 400 ms were observed for P2, and this is in accordance with previous studies [11]. The median number of days on parenteral lipid was 22 days in both groups, which is very similar to our previously published study (median: 18 days) evaluating the effect of SMOF-LE compared to S-LE on electrophysiological brain maturation, as measured by amplitude-integrated EEG [10]. Data on parenteral DHA intake were not available and therefore we estimated that the DHA intake in our study was very similar to this study (median DHA intake: 48 mg/kg/d), which is comparable to in utero transfer rates (45 mg/kg/d) (10). However, we could not confirm our previous findings where brain maturation was accelerated, measured by amplitude-integrated EEG in the SMOF-LE group (10). In the current study, a trend for accelerated maturation within the SMOF-LE group, while not significant, could be observed around term-equivalent age, and this could be related to the metabolization and synthesis of lipid for neuronal support during different stages of neonatal brain development. Complex processes taking place in early infant brain development correlate strongly with gestational age and the first year of life [35,36]. Whereas the second trimester provides a basis for neuronal circuit development, the last trimester is the deciding factor in proper cerebral communication. Special glia cells called oligodendrocytes increase in number between 15 and 20 weeks of gestational age (w GA) and differentiate and reach a peak at between 30 and 40 w GA. Once these cells are differentiated, neuronal axons undergo myelination and thereby enhance the conduction of action potentials and establish major white matter tracts. Even though the pathways involved in DHA synthesis have not bee completely resolved, besides environmental impacts and birth complications, LCPUFAs are recognized as some of the most important compounds for the synthesis of neuronal tissues [17]. Whilst normally being adequately supplied by the mother during pregnancy, preterm infants lack these needed substrates for the build-up of cell membrane composites [37]. The plasma lipid levels of DHA are low in most terrestrial animals, including humans, suggesting that the brain has particular mechanisms to concentrate DHA [38]. In vitro, glial and cerebral endothelial cells, but not neurons, can produce DHA from ALA and other precursor n-3 fatty acids [38,39]. Further understanding of brain DHA uptake will provide more insight into understanding the temporal window of brain growth and neural maturation. Recently, in a review investigating the controversies of parental lipids in preterm infants by Frazer and Martin [40], it was noted that parenteral fish oil may decrease ARA levels, which was associated with impaired growth, sepsis, and ROP. Yet, in our recent randomized trial [20], we did not find any impacts of SMOF-LE on these mentioned parameters; however, specific information on ARA levels was not collected and could not be provided in this study. The limitation of our study is the relatively small sample size and the retrospective nature, which may not be large enough to pick up a statistically significant difference. However, a potential bias of the nutritional management according to the individualized physicians’ discussion cannot ruled out. Furthermore, VEP measurements were not performed beyond term-equivalent age, as any possible significant difference at later time points has not been assessed. In this regard, Birtsch and colleagues [41] as well as Hofmann and colleagues [42] reported a positive effect of polyunsaturated fatty acid on VEP in heathy born infants, measured between birth and one year of age. Only one study, performed in 1996, looked at the effect of LC-PUFA on visual evoked potential, reporting information from birth until 52 weeks PMA [43]. In the mentioned study, a breast milk group, a LC-PUFA group rich in DHA and a control group were compared. A better wave morphology and a faster late latency component were observed in both the breast milk and the LC-PUFA group compared to the control. Due to a change in the regime of enteral nutrition that introduced human donor milk instead of formula for preterm infants until the 32nd week of gestation, more infants who received SMOF-LE than S-LE were provided with donor milk in case of the lack of their own mothers’ milk. Yet, preterm formula feeding at the time was already fortified with DHA. Prospective trials and follow-up studies are necessary in order to evaluate the potentially positive effect on visual neuronal conduction and visual pathway maturation. While SMOF-LE provides DHA in tenfold higher amounts compared to S-LE (2.2% vs. 0.2%), the DHA precursor alpha-linolenic acid is provided in 2.5 times lower amounts (SMOF-LE: 2.5 %; S-LE: 5.6 %) [44]. This might attenuate some of the potential effects of additional provision of DHA. Considering the low conversion rate from alpha-linolenic acid to DHA of only 1%, this effect is highly likely negligible [45]. However, infants in the SMOF-LE group were well-matched with infants in the S-LE group, and this study provides the first exploratory data analyzing the effect of SMOF-LE compared to S-LE on neuronal conduction from birth until discharge, as assessed by VEPs.

## 5. Conclusions

This study found no significant differences in visual neuronal conduction, measured by VEPs, in preterm infants who received a mixed-lipid emulsion containing fish oil (SMOF-LE) in comparison to a soybean oil-based lipid emulsion (S-LE) for parenteral nutrition. A possible better treatment effect in the SMOF-LE group was seen as a trend toward a shorter latency, indicating faster neural conduction at around term-equivalent age. Follow up studies are necessary to evaluate the potentially positive effect SMOF-LE on visual pathway maturation.

## Figures and Tables

**Figure 1 nutrients-13-04241-f001:**
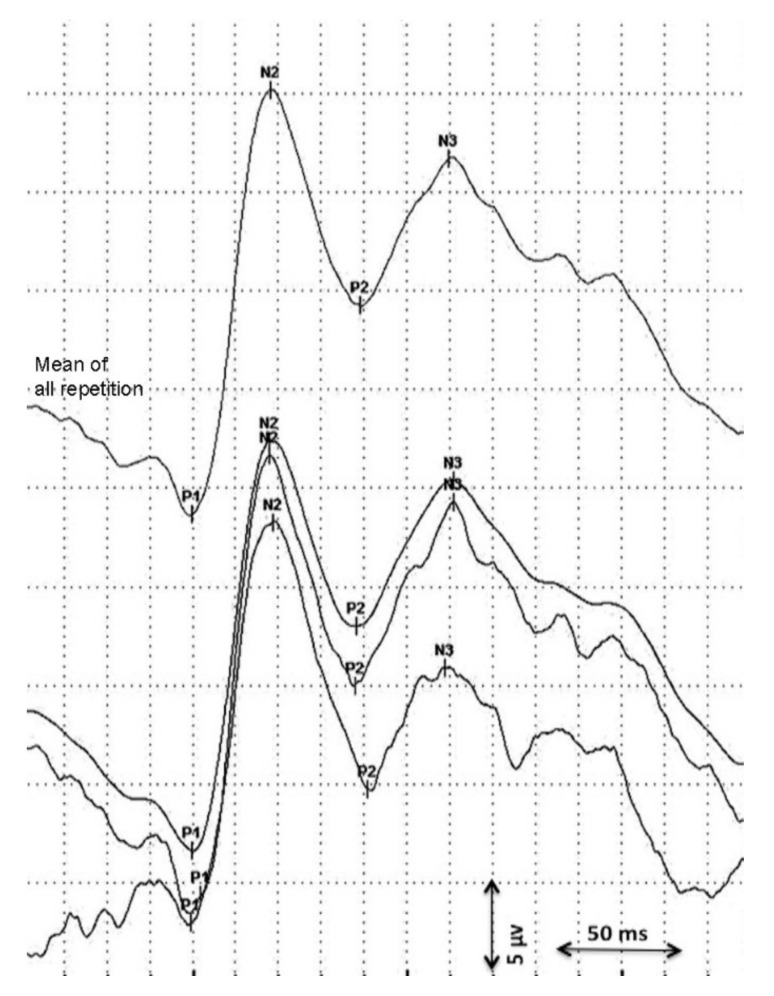
Analysis of the visual evoked potential.

**Figure 2 nutrients-13-04241-f002:**
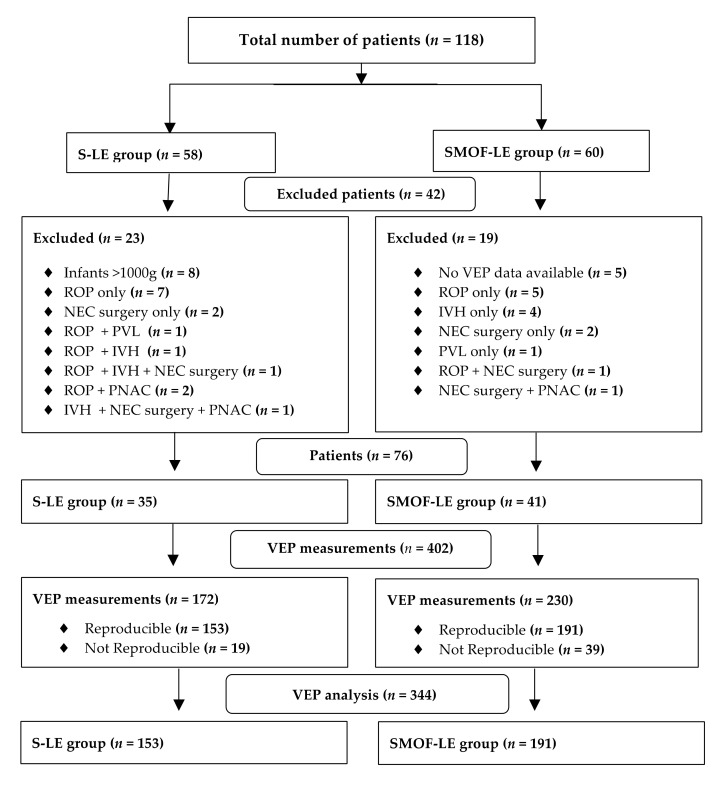
Flow diagram for patients and visual evoked potential measurement exclusion criteria.

**Figure 3 nutrients-13-04241-f003:**
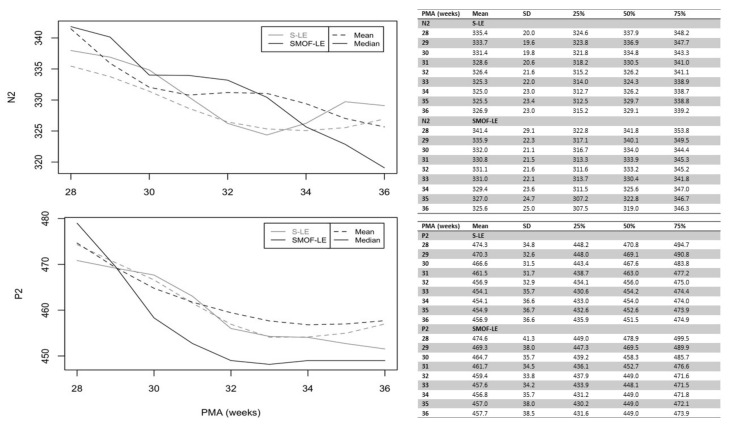
Unadjusted latencies of N2 and P2 in the S-LE and SMOF-LE groups between 28 and 36 weeks postmenstrual age (PMA).

**Figure 4 nutrients-13-04241-f004:**
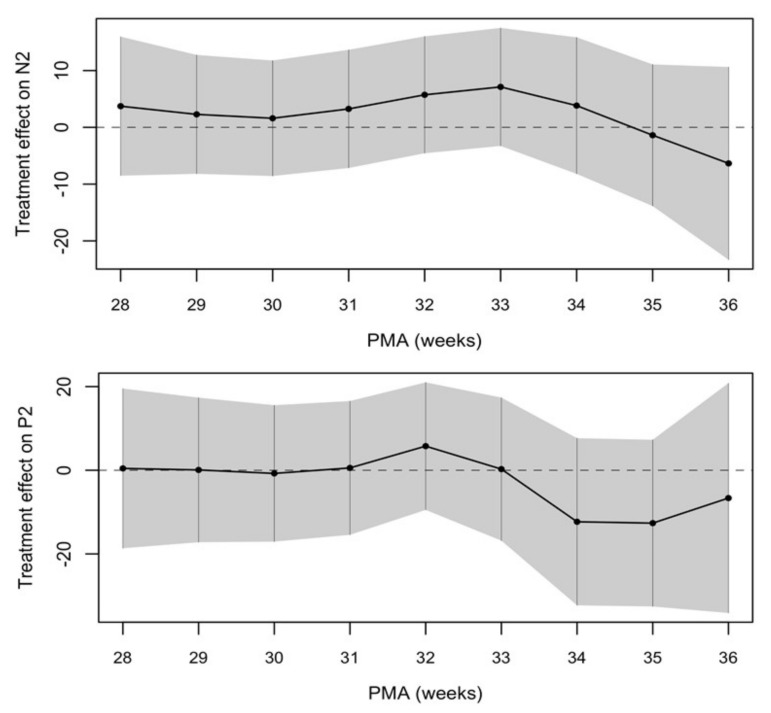
Negative treatment effect favors the SMOF-LE group at around term-equivalent age.

**Table 1 nutrients-13-04241-t001:** Descriptive characteristics.

Variables		S-LE (*n* = 35)	SMOF-LE (*n* = 41)	*p*-Values
Gender, female (*n*,%)		20 (57.1)	23 (56.1)	0.99
Gestational age, weeks (mean ± SD)		25.6 ± 1,1	25.1 ± 1.1	0.45
Birth weight, g (mean ±SD)		820 ± 173	775 ± 129	0.20
Small for gestational age (*n*,%)		0 (0)	1 (2.4)	0.99
Head circumference, cm (mean ± SD)		23.9 ± 2.2	23.2 ± 1.3	0.07
Antenatal steroids for lung maturation (*n*,%)		35 (100)	39 (95.1)	0.99
Lung maturation, complete (*n*,%)		29 (82.9)	25 (61.0)	0.09
Cesarean Section (*n*,%)		33 (94.3)	32 (78.0)	0.05
Multiple birth (*n*,%)		15 (42.9)	12 (29.3)	0.23
Umbilical artery pH (mean ± SD)		7.34 ± 0.66	7.32 ± 0.11	0.37
Apgar Score, 5 min (mean ± SD)		8.4 ± 0.6	8.5 ± 0.9	0.56
Total days of mechanical ventilation (mean ± SD)		2.0 ± 3.9	3.7 ± 6.1	0.16
ROP grade I-II (*n*,%)		17 (48.6)	23 (56.1)	0.64
IVH grade I-II (*n*,%)		7 (20.0)	5 (12.2)	0.52
PVL grade l (*n*,%)		0 (0)	1 (2.4)	0.99
Chronic lung disease (*n*,%)		6 (17.1)	9 (22.0)	0.77
NEC (*n*,%)		1 (2.9)	3 (7.3)	0.62
Total days on any parenteral nutrition (mean ± SD)		35 ± 16	31 ± 16	0.26
Total days on parenteral lipids (mean ± SD)		22 ± 10	22 ± 8	0.98
Total days on enteral nutrition (mean ± SD)		73 ± 21	86 ± 37	0.06
Weight gain per day, g (mean ± SD)		23 ± 2	23 ± 9	0.85
Feeding at discharge (*n*,%)	Human milk	10 (28.6)	15 (36.6)	0.47
	Formula	16 (45.7)	16 (39.0)	0.64
	Both	9 (25.7)	9 (22.0)	0.78
Postmenstrual age at discharge, weeks (mean ± SD)		36.2 ± 2.6	38.2 ± 5.0	0.044
Weight at discharge, g (mean ± SD)		2403 ± 560	2705 ± 1011	0.12
Weight Z-Score at discharge (mean ± SD)		−1.9 (0.6)	−1.6 (1.1)	0.13
Length at discharge, g (mean ± SD)		43.9 (4.0)	44.9 (6.8)	0.48
Length at discharge Z-Score (mean ± SD)		−0.09 (1.2)	−0.08 (0.7)	0.48
HC at discharge, g (mean ± SD)		31.8 (2.9)	32.5 (4.3)	0.43
HC Z-Score at discharge (mean ± SD)		−0.1 (0.7)	0.1 (1.1)	0.42

Retinopathy of prematurity (ROP); intraventricular hemorrhage (IVH); periventricular leukomalacia (PVL); necrotizing enterocolitis (NEC); head circumference (HC).

**Table 2 nutrients-13-04241-t002:** Estimated treatment effect on N2 and P2 between 28 and 36 weeks postmenstrual age (PMA).

**PMA (Weeks)**	**S-LE N2 Means**	**St.error**	**SMOF-LE N2 Means**	**St.error**	**Estimate**	**St.error**	**Lower 95% CI**	**Upper 95% CI**
28	334.6	3.6	338.4	4.9	3.7	6.1	−8.5	15.9
29	331.8	3.6	334.1	3.8	2.2	5.2	−8.1	12.7
30	329.0	3.6	330.6	3.5	1.6	5.9	−8.5	11.7
31	326.1	3.7	329.4	3.6	3.2	5.2	−7.1	13.6
32	324.4	3.7	330.1	3.5	5.7	5.1	−4.5	16.2
33	323.2	3.8	330.3	3.5	7.1	5.2	−3.2	17.5
34	324.3	4.5	328.2	3.9	3.8	6.1	−8.1	15.8
35	326.4	4.3	325.0	4.5	−1.3	6.2	−13.8	11.8
36	327.9	6.3	321.5	5.5	−6.3	8.4	−23.3	10.6
**PMA (Weeks)**	**S-LE P2 Means**	**St.error**	**SMOF-LE** **P2 Means**	**St.error**	**Estimate**	**St.error**	**Lower 95% CI**	**Upper 95% CI**
28	473.9	6.4	474.4	6.9	0.4	9.5	−18.6	19.5
29	470.1	6.8	470.2	6.1	0.08	8.6	−17.2	17.3
30	466.5	5.8	465.7	5.6	−0.7	8.1	−17.0	15.5
31	461.6	5.8	462.1	5.4	0.5	7.9	−15.4	16.5
32	452.1	5.7	457.8	5.6	5.7	7.6	−9.4	21.1
33	452.5	6.8	452.8	5.1	0.2	8.5	−16.8	17.3
34	464.8	8.3	452.4	5.4	−12.3	9.9	−32.2	7.6
35	463.8	7.8	451.2	6.1	−12.6	9.9	−32.5	7.2
36	462.7	8.3	456.1	11.1	−6.6	13.7	−34.1	20.8

## Data Availability

The data presented in this study are available on request from the corresponding author. The data are not publicly available due to ongoing research.

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
