# Peer review of "A Mixed-Lipid Emulsion Containing Fish Oil for the Parenteral Nutrition of Preterm Infants: No Impact on Visual Neuronal Conduction"

_nutrients, 2021, doi:10.3390/nu13124241_

Round 1
Reviewer 1 Report
This is a retrospective study on comparison of two lipid emulsion- SMOF-LE and S-LE for visual neuronal conduction in preterm infants.
Overall, although the authors did not find a significant difference in VEP in the two lipid emulsions groups, this is a novel comparison.
The authors should restate the title more reflective of the actual study and the findings. The title is a little misleading. The authors should better explain the VEP in introduction. It is unclear how the baseline VEP changes as gestational age increases. The authors need to better define the two comparison groups, this was stated as a retrospective study. Was this study where the S-LE and SMOF-LE were used in the unit at different time periods? What was the standard of practice in terms of LE in the study unit? Also, is it routine in the unit where this study was conducted to perform VEP on all ELBW infants at PMAs between 28 and 36 weeks. or was this part of a separate study? If the baseline VEP changes as the gestational age increases, then the authors should account for the change in VEP within the groups.
Although the authors do not show a statistical difference between the two groups, more studies where the findings are not overall significant should be published.
Reviewer 2 Report
The study by Binder et al. is a very nice paper about the potential of a mixed lipid emulsion for promoting visual neuronal maturation in extremely preterm infants.
I have only a few minor comments:
Methods
- the study was retrospective and performed over a quite long period. The two study periods are differentiated by a substantial change in PN policy towards lipid. Could the authors discuss if any other relevant change in parenteral and/or enteral nutrition management of EP infants took place over that period?
- could the authors explain a bit more the significance of the three peaks selected for statistical analyses?
Results
- I’m not sure to have understood the concept of “maturational features” correctly (line 155). Could you please detail the meaning? In general, a brief explanation of the significance of the measured peaks would be useful for the reader (maybe in the methods section)
Discussion
- there’s a very interesting paper about the importance of ARA, published this year in ADC by Frazer and Martin, which places the use of mixed lipid emulsions in quite a different perspective. I think it would be nice to cite this paper and discuss its proposed view on IV lipids
- the retrospective nature of the study should be acknowledged among study limitations
Reviewer 3 Report
This is a well-written report of a retrospective study evaluating the effect of mixed vs soybean lipid emulsions on visual evoked potentials in extremely low birth weight infants. The authors' verbal and visual presentation of the data were succinct and easy to follow. The study is largely limited by the small sample size and the retrospective design, both of which are acknowledged.
A few questions/points remain:
Introduction
- Would recommend adding more background discussing the utility and reliability of assessing VEPs in ELBWs, as many clinicians and clinical researchers may be unfamiliar with this type of testing in this specific population.
Methods
- In Patient Groups, please clarify that the VEP measurements were obtained for other research studies (vs clinical practice) and that this is a secondary analysis.
- What daily dose of lipids did your unit provide, and was this similar between the two cohorts / eras?
- Were there other nutritional changes that occurred during these two historical eras?
- One common concern with the use of SMOF is the decreased amount of essential fatty acids that it provides, and thus some units furnish a small dose of soybean lipids as well. Was there any additional supplementation of essential fatty acids for this group?
Results
- In Table 1, what is the p-value in the difference in human milk exposure between the two groups? Was there a difference in maternal vs donor milk exposure?
- In Table 1, the weight at discharge is not statistically different but appears clinically different. Was there a difference in length of stay and/or PMA at time of discharge? Is it possible to report the discharge weight as z-scores? Were other anthropometric measurements similar at discharge?
Discussion
- As mentioned above, SMOF has less essential fatty acids, including ALA, the precursor to DHA. I wonder if this different provision of ALA attenuated the difference in VEP between the two groups. Consider adding this to your discussion surrounding ALA.
- Would also consider strengthening your discussion that much myelination occurs after birth / term-equivalent age, again supporting that further follow up studies would be beneficial.
